# Immunization with a Multivalent *Listeria monocytogenes* Vaccine Leads to a Strong Reduction in Vertical Transmission and Cerebral Parasite Burden in Pregnant and Non-Pregnant Mice Infected with *Neospora caninum*

**DOI:** 10.3390/vaccines11010156

**Published:** 2023-01-11

**Authors:** Dennis Imhof, William Pownall, Kai Pascal Alexander Hänggeli, Camille Monney, Laura Rico-San Román, Luis-Miguel Ortega-Mora, Franck Forterre, Anna Oevermann, Andrew Hemphill

**Affiliations:** 1Department of Infectious Diseases and Pathobiology, Institute of Parasitology, Vetsuisse Faculty, University of Bern, 3012 Bern, Switzerland; 2Graduate School for Cellular and Biomedical Sciences, University of Bern, 3012 Bern, Switzerland; 3Department of Surgery, Small Animal Clinic, Vetsuisse Faculty, University of Bern, 3012 Bern, Switzerland; 4Department of Clinical Research and Veterinary Public Health, Division of Neurological Sciences, Vetsuisse Faculty, University of Bern, 3012 Bern, Switzerland; 5SALUVET, Animal Health Department, Faculty of Veterinary Sciences, Complutense University of Madrid, 28040 Madrid, Spain

**Keywords:** neosporosis, *Listeria monocytogenes* vaccine vector, multivalent vaccine, murine model

## Abstract

*Neospora caninum* is an apicomplexan parasite that causes abortion and stillbirth in cattle. We employed the pregnant neosporosis mouse model to investigate the efficacy of a modified version of the attenuated *Listeria monocytogenes* vaccine vector Lm3Dx_NcSAG1, which expresses the major *N. caninum* surface antigen SAG1. Multivalent vaccines were generated by the insertion of *gra7* and/or *rop2* genes into Lm3Dx_NcSAG1, resulting in the double mutants, Lm3Dx_NcSAG1_NcGRA7 and Lm3Dx_NcSAG1_NcROP2, and the triple mutant, Lm3Dx_NcSAG1_NcGRA7_NcROP2. Six experimental groups of female BALB/c mice were inoculated intramuscularly three times at two-week intervals with 1 × 10^7^ CFU of the respective vaccine strains. Seven days post-mating, mice were challenged by the subcutaneous injection of 1 × 10^5^
*N. caninum* NcSpain-7 tachyzoites. Non-pregnant mice, dams and their offspring were observed daily until day 25 post-partum. Immunization with Lm3Dx_NcSAG1 and Lm3Dx_NcSAG1_NcGRA7_NcROP2 resulted in 70% postnatal pup survival, whereas only 50% and 58% of pups survived in the double mutant-vaccinated groups. Almost all pups had died at the end of the experiment in the infection control. The triple mutant was the most promising vaccine candidate, providing the highest rate of protection against vertical transmission (65%) and CNS infection. Overall, integrating multiple antigens into Lm3Dx_SAG1 resulted in lower vertical transmission and enhanced protection against cerebral infection in dams and in non-pregnant mice.

## 1. Introduction

*Neospora caninum* is a cyst-forming apicomplexan parasite that is closely related to *Toxoplasma gondii*. *N. caninum* acts as the causative agent of neosporosis in cattle, dogs and small ruminants including sheep and goats [1]. Infection with *N. caninum* can lead to abortions, stillbirth and reproductive failure in several animal species, but no cases have been detected in humans [1]. *N. caninum* is economically highly significant in the dairy and beef cattle industry with financial losses amounting up to around 1.3 billion USD per year [2]. The parasite undergoes a complex life cycle comprised of three distinct infectious stages: (i) sporozoites encapsulated in oocysts, which are produced and shed into the environment by the definitive host; (ii) rapidly proliferating tachyzoites that are responsible for the acute disease and the vertical transmission and (iii) slowly dividing bradyzoites, enclosed within tissue cysts that can persist over years in the intermediate host and induce the chronic phase of the disease [1].

In human and veterinary medicine, the development and implementation of vaccines seem to be the most cost-effective strategy to eradicate infectious diseases. Several vaccine types have been evaluated against neosporosis in murine and bovine models, but there is still no vaccine available on the market [1,3]. Vaccine development against parasites with a heteroxenous life cycle is complex due to the different infectious stages, all of which exhibit distinct physiological, metabolic, and antigenic properties. An efficient vaccine against neosporosis should prevent tachyzoite proliferation and vertical transmission but should also reduce oocysts shedding from the definitive host to limit horizontal transmission. In addition, it should reduce tissue cyst formation in intermediate hosts [3,4].

In recent years, subunit vaccines such as recombinant antigens, DNA vaccines and viral and bacterial vectors containing immunodominant *Neospora* antigens have been investigated, but only with limited success [1,5,6]. According to current knowledge, live vaccination with attenuated strains appears to represent the most effective approach to protect calves from neosporosis-induced fetal death [7,8,9,10]. Nevertheless, there are several disadvantages of live-attenuated vaccines regarding safety, maintenance, production costs, stability and shelf life [3]. With killed parasite lysates, the potential risk of reverting to virulence is ruled out, but unfortunately, these types of vaccines have not been able to induce significant protection against vertical transmission in cattle [3,11]. Therefore, the search for a safe and effective vaccine is still ongoing.

*Listeria monocytogenes* is a Gram-positive, facultative intracellular bacterium that has gained much attention over recent years as a vaccine vector, due to its capacity to induce a robust innate and adaptive immune response [12,13,14,15]. After vaccination with live attenuated *L. monocytogenes*, the pathogen multiplies but is rapidly cleared, which finally results in long-lasting cell-mediated immunity mediated by antigen-specific CD8^+^ T-cells [14]. Thus, *L. monocytogenes* seems to be a suitable vaccine vector in cancer therapy and in the fight against intracellular pathogens [16,17,18]. In a previous study, we reported on the generation of the attenuated mutant *Listeria* vector Lm3Dx by deleting the three crucial virulence genes, *actA*, *inlA* and *inlB*, avoiding the systemic spread of the bacteria [19]. Additionally, Lm3Dx was engineered by inserting the gene coding for the major immunodominant surface antigen NcSAG1 into the *actA* locus, resulting in the vaccine strain Lm3Dx_NcSAG1 [19]. 

NcSAG1 is an immunodominant antigen and is postulated to play an important role in *N. caninum* host cell invasion [20]. In earlier studies, the application of a combined DNA/recombinant protein vaccine formulation (pcDNA3-NcSAG1/recNcSAG1) had resulted in partial protection against cerebral infection in non-pregnant mice [21]. We had previously shown that the *Listeria* vaccine vector Lm3Dx_NcSAG1 fulfilled several safety requirements, did not induce relevant organ damage in non-pregnant and pregnant mice, and stimulated a Th1-mediated immune response against NcSAG1 [19]. In addition, although vaccination with Lm3Dx_NcSAG1 in the pregnant neosporosis mouse model did not lead to reduced cerebral parasite burden in dams, it reduced the neonatal and postnatal mortality of newborns and also reduced the cerebral parasite burden in non-pregnant mice [18]. These promising results suggest that the protective effect could be further enhanced by inserting additional *N. caninum* antigens into Lm3Dx_NcSAG1. Promising candidates for a multivalent vaccine could include dense granule and rhoptry antigens.

Shortly following host cell invasion, apicomplexan parasites secrete dense granule (GRA) proteins into the parasitophorous vacuole, where they play essential functions [1]. For instance, Nishikawa et al. showed that an NcGRA7-knockout strain induced a significantly lower virulence in infected BALB/c mice compared to the parental Nc1 strain, and a reduced number of NcGRA7-deficient parasites could be detected in the brain. Thus, NcGRA7 appears to participate in the pathogenesis of neosporosis [22], and its role as a vaccine candidate has been studied earlier. DNA vaccines based on NcGRA7 exhibited partially protective effects against congenital neosporosis [23,24], and the immunization of BALB/c mice prior to pregnancy with a plasmid encoding NcGRA7 resulted in a reduction in vertical transmission of 54% [23]. The inclusion of the CpG adjuvant into an NcGRA7 DNA vaccine formulation resulted in considerably higher protection against congenital infection [24]. In contrast, immunization with bacterially produced recombinant NcGRA7, although inducing a very strong humoral and cellular immune response with high IFN-gamma production, did not significantly protect against congenital infection [25].

Rhoptries are unique secretory organelles, whose contents participate in the initiation of the host cell invasion processes and the formation of the parasitophorous vacuole membrane, and some rhoptry proteins also act as transcriptional regulators of the host cell [26,27]. The immunization of mice with recombinant NcROP2 emulsified in saponin adjuvant led to a significant reduction in cerebral infection, and no signs of neosporosis could be observed in vaccinated mice [28]. A recombinant vaccine formulation combining NcROP2 and NcROP40 improved pup survival in a pregnant mouse model of neosporosis [29]. Significantly higher pup survival rates and the inhibition of vertical transmission were achieved by vaccination with a cocktail of recombinant proteins composed of NcROP2, NcPDI and NcROP40, all three coupled to OprI, an outer membrane lipoprotein from *Pseudomonas aeruginosa* exerting adjuvant properties by engaging Toll-like receptor 2 [30]. These experiments demonstrated that the use of a combination of recombinant antigens, as well as the induction of a more balanced Th1/Th2 immune response through the linkage of these antigens to OprI, resulted in higher efficacy. 

The promising results obtained through the use of the Lm3Dx_NcSAG1 monovalent vaccine highlight the efficiency of the *L. monocytogenes* vector-system. Due to the promising results achieved in former studies with NcGRA7 and NcROP2 (see above), we decided to combine these three antigens into a multivalent Lm3Dx vaccine. In this study, we engineered the Lm3Dx_NcSAG1 vaccine vector to express NcSAG1 simultaneously with either NcGRA7, NcROP2, or both NcGRA7 and NcROP2, and challenged vaccinated and pregnant mice with *N. caninum* tachyzoites to comparably assess vaccine-mediated protection.

## 2. Materials and Methods

### 2.1. Host Cells, Parasite and Primers

If not indicated otherwise, all cell culture media and materials were purchased from Gibco-BRL (Zürich, Switzerland). Human foreskin fibroblasts (HFF; ATCC^®^ SCRC-1041^TM^) were cultured in Dulbecco’s modified Eagles’ medium (DMEM) supplemented with 10% fetal calf serum (FCS) and 1% antibiotics/antimycotics at 37 °C, 5% CO_2_ in T25, T75 or T175 cell culture flasks (Sarstedt, Sevelen, Switzerland). BALB/c dermal fibroblasts (CELLNTEC Advanced Cell Systems AG, Bern, Switzerland) were maintained under the same conditions as described above. HFF cells were infected with the highly virulent *N. caninum* strain NcSpain-7 strain and maintained for several passages before parasites were transferred to BALB/c dermal fibroblasts, three days prior to the challenge of mice [31].

### 2.2. Generation of Different Attenuated Mutant Listeria Strains

The generation of the Lm3Dx_NcSAG1 strain was already described in our previous study [19]. Briefly, the JF5203∆actA mutant was used as a parental strain to create the vaccine vector JF5203∆actA/inlA/inlB/fosX (Lm3Dx) by the in-frame deletion of the three genes by homologous recombination. Lm3Dx_NcSAG1 was created by electroporating the plasmid pMAD_NactA100AA_SAG1 comprehending the NcSAG1 gene in Lm3Dx. A sequence of selection on plate and temperature changes as described [19] was performed to insert this gene at the locus of the previously removed *actA* Lm virulence gene. This strain was used as the parental strain for the development of Lm3DX_NcSAG1_NcGRA7, Lm3Dx_NcSAG1_NcROP2 and Lm3Dx_NcSAG1_NcGRA7_NcROP2. The plasmids pMAD_NactA100AA_NcGRA7, pMAD_NactA100AA_NcROP2 and pMAD_NactA100AA_NcGRA7_NcROP2 were created in silico using the Geneious 8.1 software (Biomatters Inc., Auckland, New Zealand) and ordered at Twist Bioscience (San Francisco, CA, USA). Briefly, the *gra7* GenBank: AAC47661.1 and the *rop2* GenBank: ADM48813.1 were codon-optimized for *L. monocytogenes* using publicly available web-based software (http://www.jcat.de/; accessed on 15 October 2021). Similar to Lm3Dx_NcSAG1, the first 300 nucleotides of *actA* were added at the 5′ end of each sequence. To ensure insertion of NcGRA7 and NcROP2 or both in the proper locus, the same flanking regions of *inlAB* used in our previous study were added at the 5′ and 3′ end of the sequence we desired to introduce into our vaccine. Finally, Lm3Dx_NcSAG1 was transformed with one of the following plasmids: pMAD_NactA100AA_NcGRA7, pMAD_NactA100AA_NcROP2 and pMAD_NactA100AA_NcGRA7_NcROP2 to create Lm3Dx_NcSAG1_NcGRA7, Lm3Dx_NcSAG1_NcROP2 and Lm3Dx_NcSAG1_NcGRA7_NcROP2 with NcGRA7 and/or NcROP2 in the *inlAB* locus under the control of the *actA* promotor. The protocol of Arnaud et al. was used for homologous recombination similarly as in our previous study [19,32]. Proper insertion was confirmed by DNA sequencing of the resulting mutants.

### 2.3. Ethics Statement

All protocols involving animals were approved by the Animal Welfare Committee of the Canton of Bern under the license BE113/19. Mice were purchased from a commercial breeder (Charles River, Sulzberg, Germany) and were maintained in a common room under controlled temperature with 14 h/10 h light and dark cycles, with food and water accessible ad libitum according to the guidelines of the Animal Welfare Legislation of the Swiss Veterinary Office. Animals were handled in strict accordance with guidelines to minimize suffering.

### 2.4. Efficacy Assessment of the Different Mutant Listeria Strains in Pregnant and Non-Pregnant Mice Infected with NcSpain-7 Tachyzoites

For the efficacy evaluation of the different vaccine strains, 96 female and 48 male BALB/c mice, 8 weeks of age, were used. After an acclimatization period of two weeks, female mice were randomly distributed into six experimental groups that consisted of 16 mice, with 2 females per cage. The experimental groups are listed in Table 1. The immunization schedule and the timepoints of mating and blood and organ collection are depicted in Figure 1. 

The inoculation doses of each vector were prepared as described [19]. Briefly, after culturing a colony overnight in 10 mL BHI broth at 37 °C, bacteria were washed 3 times with sterile PBS and were diluted to achieve a concentration of 1 × 10^7^ CFU of the respective mutant *L. monocytogenes* vaccine strain in 50 µL. The concentration was confirmed by the plating of the inoculum on BHI-Agar plates (Sigma-Aldrich) and the determination of the CFU 24 h after incubation at 37 °C. The mice were then immunized with 1 × 10^7^ CFU of the respective mutant *L. monocytogenes* vaccine strain, diluted in 50 µL of PBS (different strains are indicated in Table 1) by intramuscular injection in the thigh, three times in total at two-week intervals. The infection control group (C+) was inoculated with the empty vector Lm3Dx, whereas the negative control (C−) only received phosphate-buffered saline (PBS). Eight days after the second immunization, mice were oestrus-synchronized by the Whitten effect [33] for three days. Thereafter, two female mice were housed together with one male for 72 h. The third vaccination was applied directly after the separation of the males from the females, corresponding to four days before challenge. Three days prior to infection, NcSpain-7 tachyzoites were transferred from human foreskin fibroblast (HFF) monolayers to BALB/c dermal fibroblasts and were maintained at 37 °C, 5% CO_2_ until they were further processed for challenge on day 32 [30]. Prior to infection, NcSpain-7 tachyzoites were harvested from cell culture flasks. Counting and calculations of tachyzoites were performed as described earlier [34]. All mice, except the negative control (C−), were infected with a sublethal dose of 1 × 10^5^ tachyzoites of the highly virulent *N. caninum* strain NcSpain-7 by subcutaneous injection in the neck. Mice of the C− group were injected only with 1 × 10^5^ BALB/c dermal fibroblasts. After challenge, the mice were observed daily for clinical signs of neosporosis, and the weights were measured every third day to confirm pregnancy and/or to detect possible pregnancy losses. At day 18 post-mating, pregnant mice were transferred to single cages, whereas non-pregnant mice were maintained in groups of 3–4 animals. The pregnant mice gave birth between days 20 and 22 of pregnancy. Data on the clinical signs, fertility, litter size and neonatal and postnatal mortality were recorded daily. Between days 70 and 72, corresponding to 25 days post-partum (p.p.), non-pregnant mice, dams and pups were euthanized using a chamber filled with isoflurane and CO_2_. Blood was directly collected after euthanasia by cardiac puncture, and serum was stored at −20 °C after centrifugation at 1200× *g* for 12 min at 4 °C. Brain samples were collected aseptically, and one hemisphere of each animal was stored at −20 °C for quantitative real-time PCR, whereas the other hemisphere was fixed in 10% formalin and further processed for histological analysis.

### 2.5. Evaluation of Cerebral Parasite Burden by Quantitative Real-Time (RT) qPCR

Cerebral parasite burdens were analyzed in non-pregnant mice, dams and surviving pups by RT-qPCR established for *N. caninum* [18,35,36]. Pups which died before the end of the study were considered *N. caninum*-positive. DNA purification from brain tissues was performed with the NucleoSpin DNA RapidLyse Kit (Macherey-Nagel, Oesingen, Switzerland) according to the manufacturer’s instructions. Afterwards, DNA concentration was quantified by using the QuantiFluor double-stranded DNA system (Promega, Madison, WI, USA). TaqMan probe-based RT-qPCR was performed in a CFX96 qPCR instrument (Bio-Rad Laboratories AG, Cressier, Switzerland) to quantify *N. caninum* DNA from brain samples. CFX manager software version 1.6 was used for the analysis of the PCR results. RT-qPCR is targeted to the repetitive genomic sequence Nc5 of *N. caninum* [35]. The *N. caninum* PCR reaction was prepared as previously described [18]. In brief, the reaction mixture (10 μL per reaction) contains 5 μL of 2× Mastermix (SensiFASTTM Probe NO-ROX Kit; Bioline Meridian Lifescience, Memphis, TN, USA), 500 nM forward primer Np21plus (5′-CCCAGTGCGTCCAATCCTGTAAC-3′) and reverse primer Np6plus (5′-CTCGCCAGTCAACCTACGTCTTCT-3′) [35], 100 nM of detection probe NC5-1 (5′-*FAM*-CACGTATCCCACCTCTCACCGCTACCA-*BHQ-1*-3′) [36] and 5 ng of sample DNA. Additionally, 300 nM dUTP (supplementary to dTTP included in the 2× Mastermix) and 1 unit of heat-labile Uracil DNA Glycosylase (UDG) (both from Bioline Meridian Lifesciences) were included in the reaction mixture to remove eventual carry-over contaminations from previous reactions as described earlier [37]. For UDG-mediated decontaminations, the temperature profile included an initial 10 min incubation at 40 °C followed by a 5 min denaturation period at 95 °C. Subsequently, DNA amplification was accomplished during 50 cycles of 10 s at 95 °C and 30 s at 60 °C. After each cycle, light emission by the fluorophore was measured at 60 °C. Brain samples from adult mice were tested in duplicates, whereas pup brains were measured as single values [18]. As an external standard, samples containing DNA equivalents from approximately 10,000, 1000, 100 and 10 *N. caninum* NcSpain-7 tachyzoites were included in each PCR run. Probes and primers for quantitative real-time PCR were purchased from Microsynth (Balgach, Switzerland).

### 2.6. Histological Evaluation of Cerebral Tissue and Encephalitic Grade Assessment

One brain hemisphere of all non-pregnant and pregnant mice of the Lm3Dx_NcSAG1, Lm3Dx_NcSAG1_NcGRA7_NcROP2 and the control groups was fixed in 10% formalin. Afterwards, fixed brain specimens were embedded in paraffin, cut at 4 µm and stained with hematoxylin and eosin (H&E). *N. caninum*-associated lesions were assessed under the light microscope and encephalitic grades were scored from 0–3 (from not affected to severely affected brain sections) as described before [18].

### 2.7. Assessment of Antibody Responses by NcGRA2 and NcROP2 ELISA

Enzyme-linked immunosorbent assay (ELISA) was used to evaluate IgG antibody titers against NcSAG1 and *N. caninum* crude extract (from NcSpain-7 tachyzoites) in sera of non-pregnant and pregnant mice [28,38]. In short, 96-well plates (Sarstedt, Sevelen, Switzerland) were coated either with 100 ng/well recNcSAG1, recNcGRA7, recNcROP2 or with 100 ng/well of an NcSpain-7 soluble protein extract diluted in coating buffer (50 mM sodium bicarbonate; 50 mM sodium carbonate dissolved in demineralized water; pH 9.6). Plates were incubated overnight at 4 °C. The next day, plates were washed twice with washing buffer (0.05% PBS-Tween-20) before non-specific binding sites were blocked with blocking buffer (1% bovine serum albumin dissolved in 0.05% PBS-Tween-20) for at least 2 h at room temperature. Each serum sample was diluted 2-fold in blocking solution to prevent saturation before samples were added and incubated for 90 min at room temperature. After three more washing steps, the secondary antibody (anti-mouse IgG (H&L) conjugated with alkaline phosphatase (Promega, Madison, WI, USA) was diluted and applied for 1 h at room temperature. Alkaline phosphatase substrate (1 mg phosphatase substrate dissolved in 1 mL AP-substrate buffer (1 M diethanolamine; 0.01% magnesium chloride hexahydrate) was prepared and added to develop the enzymatic reaction for 25 min before absorbance was measured as optical density (OD) at 405 nm in a microplate reader (Hidex Sense plate reader, Turku, Finland). The same positive and negative control samples were used for each plate to compare OD values between samples measured in different plates. Every OD value was converted into a relative index per cent (RIPC) value by the following formula [RIPC = (OD_450nm_ sample—OD_450nm_ negative control)/(OD_405nm_ positive control—OD_405nm_ negative control) × 100] [31].

### 2.8. Detection of Antibody Responses to NcROP2 by Western Blotting

Serum samples were analyzed for the presence of antibodies directed against NcROP2 by the separation of recombinant NcROP2 (100 µg) by SDS-PAGE followed by Western blotting according to standard procedures [30]. The sera were incubated at 1:100, and an anti-mouse IgG antibody conjugated to alkaline phosphatase (Promega) at 1:2000 dilution was employed.

### 2.9. Statistical Analysis

The statistical analysis of neonatal and postnatal mortality rates as well as vertical transmission rates has been evaluated between the distinct vaccine groups and the positive control group by a Chi-squared (and Fisher’s exact) test. Cerebral parasite load, IgG titers and encephalitic grades were compared between groups by the non-parametric Kruskal–Wallis test, including Dunn’s multiple comparison test. If statistical differences between groups were detected, a Mann–Whitney U test was subsequently applied comparing only two groups with each other. Pup mortality rates were compared by plotting survival events at each time point in Kaplan–Meier graphs and afterwards, survival curves were compared by the Log-Rank (Mantel–Cox) test. The GraphPad Prism software version 9.4.0 for macOS was used to conduct all statistical analyses (GraphPad Software, La Jolla, CA, USA, www.graphpad.com accessed on 3 June 2022).

## 3. Results

### 3.1. Safety and Efficacy of the Different L. monocytogenes Vaccine Strains

Information on the strains used for vaccination, the different experimental groups and the dosages used for vaccination as well as infection is presented in Table 1. The results on the safety and efficacy parameters of the different *L. monocytogenes* vaccine strains are summarized in Table 2. The safety aspects of the attenuated *Listeria* strain Lm3Dx_NcSAG1 have been described in detail in a previous publication [19]. For the current study, three new vaccine strains were engineered: the two double mutants Lm3Dx_NcSAG1_NcGRA7 and Lm3Dx_NcSAG1_NcROP2 as well as the triple mutant Lm3Dx_NcSAG1_NcGRA7_NcROP2. At a dosage of 1 × 10^7^ CFU, the three newly engineered strains did not affect reproductive parameters such as litter size and fertility rates, similar to what had been shown for Lm3Dx_NcSAG1 (Table 2). Two out of five dams from the infection control group C+ showed clinical signs of neosporosis. In addition, one dam of the group that was vaccinated thrice with Lm3Dx_NcSAG1 exhibited a rough coat. All the other dams from the remaining experimental groups did not exhibit clinical signs. Non-pregnant mice showed no clinical signs and remained healthy until the end of the study. Vaccination with Lm3Dx_NcSAG1 and Lm3Dx_NcSAG1_NcGRA7_NcROP2 resulted in 70% pup survival, whereas in the groups that were inoculated with the double mutants (Lm3Dx_NcSAG1_NcGRA7 and Lm3Dx_NcSAG1_NcROP2), 50% and 58% of pups survived, respectively. In the control group vaccinated with the empty vector, only 7% of pups survived (56 out of 60 pups died until the end of the experiment). In contrast, only two out of 34 pups died in the negative control group, which resulted in a pup survival rate of 94%. The two pups died directly after delivery, most probably due to birth complications (Figure 2, Table 2).

### 3.2. Evaluation of the Cerebral Parasite Burden and Vertical Transmission Rates

The cerebral parasite load and vertical transmission rates were determined using RT-qPCR. Five out of nine brain tissues from dams immunized with Lm3Dx_NcSAG1, three out of six brains from dams vaccinated with Lm3Dx_NcSAG1_NcGRA7, two out of nine brain samples from dams vaccinated with Lm3Dx_NcSAG1_NcROP2 and two out of eight brains from mice vaccinated with the triple vaccine strain Lm3Dx_NcSAG1_NcGRA7_NcROP2 were tested PCR-positive for *N. caninum*. In contrast, all brain samples in the positive control group were *N. caninum* PCR-positive, whereas no parasite DNA was detected in the brain tissues of the negative control group (Table 2). The pregnant mice that had been vaccinated three times either with the two double mutants or the triple mutant exhibited a significantly lower cerebral parasite burden compared to the positive control (*** p* < 0.0084, ** p* < 0.0356; Mann–Whitney U test) (Figure 3A). 

In non-pregnant mice vaccinated with Lm3Dx_NcSAG1, two out of seven mice were positive for *N. caninum*, whereas in the two double mutants, four out of ten and three out of seven brain samples tested positive (Lm3Dx_NcSAG1_NcGRA7 and Lm3Dx_NcSAG1_NcROP2, respectively). Only one brain sample out of eight was PCR-positive in non-pregnant mice that were immunized with the triple vaccine formulation Lm3Dx_NcSAG1_NcGRA7_NcROP2. Six out of seven brain tissue samples were PCR-positive in the positive control group, and no *N. caninum* DNA could be detected in the negative control group (Table 2). Statistically significant reductions in parasite load were achieved by vaccinating non-pregnant mice either with the double mutant Lm3Dx_NcSAG1_NcGRA7 or with the triple mutant Lm3Dx_NcSAG1_NcGRA7_NcROP2 (*** p* < 0.0078, ** p* < 0.0275; Mann–Whitney U test) (Figure 3B). 

The vertical transmission of *N. caninum* tachyzoites was clearly inhibited in dams vaccinated with the triple mutant, resulting in 65% of pups testing PCR-negative. In mice vaccinated with the two double mutants Lm3Dx_NcSAG1_NcGRA7 and Lm3Dx_NcSAG1_NcROP2, 47% and 55% of pups tested negative for *N. caninum*, respectively. Furthermore, 52% of pups were PCR-negative in the Lm3Dx_NcSAG1 group. A vertical transmission rate of 95% was achieved in the positive control group, whereas all pups from the negative control tested negative (Table 2). Overall, immunization with the Lm3Dx_NcSAG1_NcGRA7_NcROP2 vector was the most effective leading to a strong reduction in vertical transmission and to increased pup survival. In addition, Lm3Dx_NcSAG1_NcGRA7_NcROP2 vaccination led to a significant decrease in the cerebral parasite burden in the non-pregnant mice and dams.

### 3.3. Histological Analysis of Cerebral Tissues

Formalin-fixed and paraffin-embedded brain tissue samples of dams and non-pregnant mice from the following groups were further assessed by histopathological analysis: the C− control group, the C+ control group, the Lm3Dx_NcSAG1 group and the Lm3Dx__NcSAG1_NcGRA7_NcROP2 group. A grading of encephalitic lesions was performed, the results of which are shown in Figure 4. Encephalitic lesions consisted of lymphohistiocytic meningitis and encephalitis with gliosis and sometimes distinct and large granulomas with central necrosis and mineralization. In a few animals, protozoa were observed in the H&E brain sections. Only three and six brains of the mice immunized with the triple mutant and with Lm3Dx_SAG1, respectively, contained encephalitic lesions. In comparison, 13 animals in the C+ group vaccinated with the empty vector Lm3Dx and infected with *N. caninum* tachyzoites exhibited lesions of varying severity.

### 3.4. Assessment of Humoral Immune Response Induced by Distinct L. monocytogenes Vaccine Strains

Sera were collected from all dams and non-pregnant mice at the endpoint (25 days post-partum), and IgG responses were assessed by ELISA using plates coated with *N. caninum* extract (Figure 5A). Overall, IgG levels were rather variable between the individual animals. The IgG levels were highest in the group C+ treated with Lm3Dx and infected with *N. caninum* in both non-pregnant and pregnant mice, whereas the IgG levels were clearly lower in all other groups. Recombinant NcSAG1, NcGRA7 and NcROP2 were expressed in *E. coli* and purified by Ni^2+^-affinity chromatography. ELISA detected IgG responses against recombinant NcSAG1 (Figure 5B) and recombinant NcGRA7 (5C), whereas anti-NcROP2 IgG could not be detected by ELISA (data not shown). With regard to NcSAG1, the highest IgG levels were measured in sera from non-pregnant mice vaccinated with Lm3Dx_SAG1, whereas IgG levels were lower in the sera obtained from mice immunized with the double mutants, and these differences were statistically significant (*** p <* 0.002 for Lm3Dx_NcSAG1_NcGRA7 and *** p <* 0.007 for Lm3Dx_NcSAG1_ROP2; Mann–Whitney U), but this was not observed in sera from pregnant mice (Figure 5B). For NcGRA7, the antibody levels in the non-pregnant mice were the highest in the groups vaccinated with Lm3Dx_NcSAG1_NcGRA7 and the triple mutant (Figure 5C). The IgG levels of Lm3Dx_NcSAG1_NcGRA7 were significantly elevated compared to C+ (*** p* < 0.0046; Mann–Whitney U). In dams, the IgG antibodies generated due to the immunization with Lm3Dx_NcSAG1_NcGRA7 followed by infection were also slightly increased in comparison with the non-vaccinated C+ group, but the differences were not significant. 

In order to further verify whether anti-ROP2 antibodies had been generated, recombinant NcROP2 was assessed by Western blotting employing serum samples of mice vaccinated with Lm3Dx_NcSAG1_NcGRA7_NcROP2 (Figure 6). This showed that antibodies were produced at various amounts in the different mice. In non-pregnant animals, four out of eight mice exhibited ROPs staining, two with strong signals, and two only weakly positive. From the eight serum samples from the dams, three exhibited ROP2 labeling, although at a lower intensity (Figure 6B). 

## 4. Discussion

Vaccine development against apicomplexan parasites is inherently difficult due to the biological complexity and the different developmental stages that form the life cycle of these organisms [39]. The requirements for an efficacious vaccine against neosporosis are severalfold, as it must be able to stimulate cytotoxic T-cell responses and T-helper cell responses and must also activate B-cell responses. In addition, these activities are dependent on how an innate immune response is modulated, and how antigen-presenting cells are activated influences the ultimate outcome of the acquired immune response [40]. In recent years, several research groups have provided evidence that *L. monocytogenes* modulates innate immunity, stimulates antigen-specific cellular responses and is highly suitable for genetic engineering [13,14,41,42]. As attenuated *L. monocytogenes* can activate CD8^+^ T-cells and CD4^+^ T cells, and as *N. caninum* occupies an intracellular niche, it is conceivable that this parasite is highly susceptible to cytotoxic T-cell responses. Therefore, we hypothesized that an *L. monocytogenes* vaccine vector expressing immunogenic parasite antigens that are involved in host–parasite interactions could be applied to induce protective immune responses for the prevention of neosporosis [13,19].

Besides NcSAG1, which represents the major immunodominant surface antigen of *N. caninum* tachyzoites, two other antigens, namely, NcGRA7 and NcROP2, were introduced into Lm3Dx, yielding two double mutants expressing NcSAG1/NcGRA7 and NcSAG1/NcROP2, as well as a triple mutant expressing NcSAG1/NcGRA7/NcROP2 in combination. All three antigens are functionally implicated in the processes leading to host cell invasion and intracellular host–parasite interaction [43]. For instance, NcSAG1 is constitutively expressed on the tachyzoite surface and is likely to mediate the initial contact between the parasite and the host cell surface membrane. NcSAG1 has been previously assessed as a vaccine candidate by employing different expression systems such as DNA-vaccines and recombinant antigens [21], the integration of a recombinant antigen into immune-stimulating complexes (ISCOMs) [44] and expression in silkworm larvae (*Bombyx mori*) using nucleopolyhedrovirus (BmNPV) bacmid expression systems [45] or Rous sarcoma virus-like particles (RSV-LP) displaying both NcSAG1 and NcSRS2 [46]. NcROP2 has also been shown earlier to represent a promising vaccine candidate, either alone or in combination with other recombinant antigens [38]. Upon host cell invasion by tachyzoites, the protein is associated with the nascent parasitophorous vacuole membrane (PVM), vacuoles surrounding the host cell nucleus and, in some instances, the surface of intracellular parasites. NcROP2 was also detected on the surface of extracellular parasites entering the host cells, and antibodies directed against NcROP2-specific peptides partially neutralized invasion in vitro [47]. Dense granule antigens, such as NcGRA7, are secreted during or shortly after the completion of host cell invasion, and most of them are involved in the modification of the parasitophorous vacuole, the vacuolar tubular network and the PVM. Several dense granule antigens are further transported to the host cell cytoplasm and the nucleus, where they act as regulators of signaling pathways [48], whereas others affect the transport through the vacuolar membrane [49]. Previous studies have shown that the immunization of mice with plasmid DNA coding for NcGRA7 conferred partial protection against the vertical transmission of *N. caninum* [23]. NcGRA7 has been postulated to exhibit immunomodulatory properties [22], and the loss of NcGRA7 triggers an inflammatory response in the placenta, resulting in the decreased vertical transmission of *N. caninum* in mice [50]. Recently, it was shown that *N. caninum* tachyzoites lacking GRA7 expression exhibited pronounced lower virulence compared to wild-type parasites [51].

Similar to earlier investigations employing Lm3Dx_NcSAG1 [18,19], we did not find any indication that the newly generated multivalent *L. monocytogenes* vaccine strains expressing multiple antigens cause safety issues. Neither the double nor the triple mutant had any impact on fertility rates, pregnancy outcomes or litter sizes. No signs of listeriosis were observed in any of the animals, and neonatal pup mortality in the vaccinated groups was not significantly increased compared to the negative control.

In terms of pup survival, both Lm3Dx_NcSAG1 and the triple mutant Lm3Dx_NcSAG_NcGRA7_NcROP2 exhibited superior efficacy, with 70% surviving pups, confirming the earlier results of Lm3Dx_NcSAG1 [18]. In contrast to the earlier study, however, the neonatal mortality in the Lm3Dx_NcSAG1 group was slightly increased, but the postnatal mortality was lower (25%) compared to earlier (33%), and the vertical transmission rate (48%) was higher in this study compared to the previous study (39%) [18]. These variations can be explained by subtle differences in the virulence of the respective inoculum. Similarly, to other apicomplexans, the infectivity of *N. caninum* tachyzoites derived from tissue culture is dependent on the number of passages these parasites have undergone in vitro, as well as the number of passages through mice [52]. Other factors could be related to the number of passages of the host cells that might also impact on parasite virulence, the handling of parasites prior to infection and yet unexplored effects obtained upon the extracellular maintenance of tachyzoites during a limited time frame [53].

Whereas the introduction of the two additional *N. caninum* antigens NcGRA7 and NcROP2 into the Lm3Dx_NcSAG1 vector did not have a notable effect on pup survival, a profound impact on vertical transmission was noted: vaccination with the trivalent Lm3Dx_NcSAG1_NcROP2_NcGRA7 was the most efficient, resulting in 65% inhibition of vertical transmission, whereas 52% of the pups from Lm3Dx_NcSAG1-vaccinated dams tested PCR negative, and the protection levels of the two double mutants ranged from 47–55%. Thus, introducing all three antigens into the Lm3Dx vaccine vector had an important impact on vertical transmission, but did not notably affect pup survival.

In our experiments, the infection of dams and non-pregnant mice was carried out with an infection dose that would normally not induce severe clinical symptoms [34]. Nevertheless, parasite DNA was identified in the brain tissues of all non-vaccinated dams and in six out of seven non-pregnant mice at the end of the experiment. In vaccinated dams and non-pregnant mice, however, the number of PCR-positive brain tissue samples were clearly reduced in all groups and most profoundly in the groups vaccinated with Lm3Dx_NcSAG1_Nc-GRA7_NcROP2. Analysis of the cerebral parasite load in dams and non-pregnant mice showed that, in general, CNS infection was less pronounced in non-pregnant mice, which had not undergone pregnancy-associated immunomodulation. A significant reduction in parasite load compared to the control group was seen in all vaccinated dams, except the one group immunized with Lm3Dx_NcSAG1, which mirrors previous results [18]. In non-pregnant mice, however, the results were different in that immunization with the double mutant Lm3Dx_NcSAG1_NcGRA7 as well as the triple mutant Lm3Dx_NcSAG1_NcGRA7_NcROP2 resulted in significantly reduced cerebral parasite loads, but neither immunization with Lm3Dx_NcSAG1 nor Lm3Dx_NcSAG1_NcROP2 had an effect. Vaccination also had an impact on the degree of encephalitis in brains, as investigated by histopathology. The encephalitic grading of the formalin-fixed brain tissues of the mice showed that in groups vaccinated with Lm3Dx_NcSAG1_NcGRA7_NcROP2 and Lm3Dx_NcSAG1, encephalitic lesion scores were significantly reduced in comparison to the group vaccinated with the empty vector.

Although we show here that generating a Lm3Dx vaccine vector that expresses more than one antigen results in improved protection, it remains unclear which of these antigens is most effective in inducing protective immunity. Analysis of the humoral immune responses against whole *N. caninum* antigen extract in sera collected at the endpoint showed that the IgG levels could be correlated to the protective effects; thus, lower IgG levels mirrored lower parasite loads, especially for the mice vaccinated with Lm3Dx_NcSAG1_NcGRA7_NcROP2. This tendency was less evident when performing NcSAG1 and NcGRA7 ELISAs. It is possible that the native and recombinant antigens do not display the same epitopes. However, further investigations are necessary to clarify this aspect. Specific anti-ROP2 antibodies could not be detected by ELISA, which was not surprising, as this has been previously reported by Aguado-Martinez et al. (2019) upon the vaccination of mice with OprI-conjugated ROP2 [30]. However, Western blot analysis showed that at least some mice vaccinated with Lm3Dx_NcSAG1_NcROP2 had generated detectable antibodies against NcROP2. Although NcROP2 has been investigated as a vaccine candidate in various studies, it is not known as an immunodominant antigen, at least with respect to the humoral immune response. Overall, the role of all antigens presented here in the cellular immune response requires further investigation.

Collectively, the efficacy results obtained in this study mark an improvement to the previous study using the monovalent Lm3Dx_NcSAG1 vaccine vector [18], especially with regard to increased protection against vertical transmission, and the improved protection against brain infection in adult mice. The inclusion of NcSAG1, NcGRA7 and NcROP2 in this study was almost as efficacious as a recently employed vaccine-linked chemotherapy approach employing Lm3Dx_NcSAG1 in combination with treatment with the bumped kinase inhibitor BKI-1748 [54]. In fact, the triple mutant *L. monocytogenes* vaccine was only slightly less efficient in preventing vertical transmission (65% versus 77%), but more efficient in terms of inhibiting brain infection in non-pregnant adult mice compared to the vaccine–drug combination. Future studies will focus on the identification and assessment of other targets for vaccination. The Lm3Dx vaccine vector represents a versatile tool to carry out such investigations, not only for the prevention of neosporosis but potentially also for other diseases caused by apicomplexan parasites.

## 5. Conclusions

In conclusion, we have demonstrated that *L. monocytogenes*-derived multivalent vaccines generated by the insertion of *gra7* and/or *rop2* genes into the Lm3Dx_NcSAG1 vaccine vector conferred increased efficacy compared to vaccination with the monovalent Lm3Dx_NcSAG1 vaccine. The triple mutant was the most promising vaccine candidate providing the highest rate of protection against vertical transmission and CNS infection. Overall, integrating multiple antigens into Lm3Dx_SAG1 represents a promising avenue to reduce the vertical transmission of *N. caninum*, and to enhance protection against cerebral infection in dams and in non-pregnant mice, and further studies should be undertaken to elucidate the immune parameters that lead to increased protectivity. In addition, one way forward should be the assessment of this approach in the more relevant ruminant target hosts. 

## Figures and Tables

**Figure 1 vaccines-11-00156-f001:**
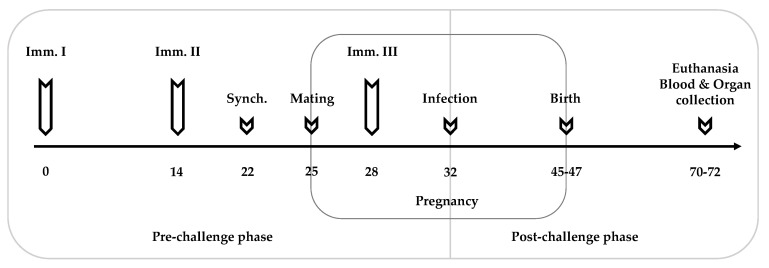
Experimental design. Mice were vaccinated three times (Imm. I–III) with the appropriate vaccine strain at two-week intervals (see Table 1). Three days before mating, the mice were oestrus-synchronized (Synch.) by the Whitten effect before two females were put together with one male. Seven days post-mating, corresponding to the first trimester of pregnancy, all mice, except the negative control, were infected with a sublethal dose of 1 × 10^5^ NcSpain-7 tachyzoites. The mice were observed daily for clinical signs. Non-pregnant mice, dams and pups were euthanized between days 70–72, corresponding to 25 days p.p.

**Figure 2 vaccines-11-00156-f002:**
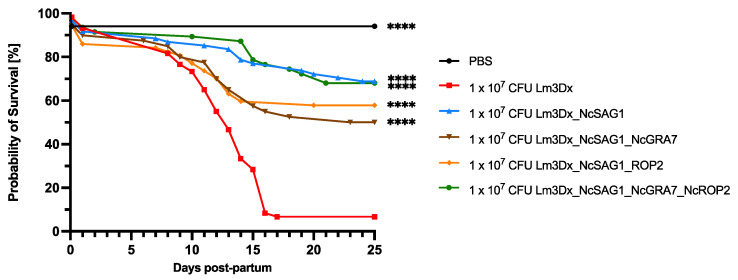
Survival curves of the different *L. monocytogenes* vaccine strains in the pregnant neosporosis mouse model. Survival rates of pups were plotted daily in Kaplan–Meier graphs and each curved was analyzed by the Log-rank (Mantel–Cox) test. Significant differences were calculated by comparing the different strains with the positive control group (**** *p* < 0.0001).

**Figure 3 vaccines-11-00156-f003:**
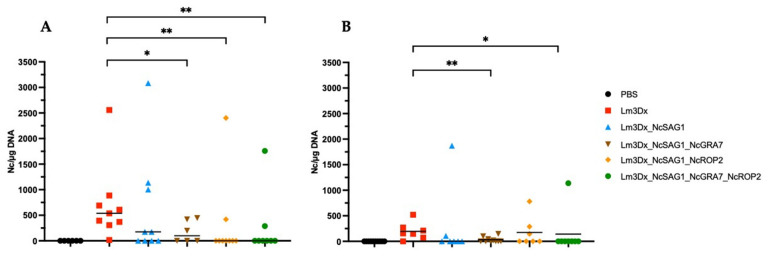
Evaluation of cerebral parasite load in dams (**A**) and non-pregnant mice (**B**). Directly after euthanasia, brain tissues were collected aseptically and further processed to quantify *N. caninum* DNA (Nc/µg DNA) by RT-qPCR. Values are arranged in scatter plots. Dams that were immunized thrice either with the double mutants or the triple mutant vaccine showed statistically significant differences compared to the positive control group, which was inoculated thrice with the empty vector Lm3Dx (*** p* < 0.0084, ** p* < 0.0356; Mann–Whitney U test). In non-pregnant mice, statistically significant reductions in cerebral parasite burden were achieved by inoculating animals with the double mutant Lm3Dx_NcSAG1_NcGRA7 or with the triple mutant Lm3Dx_NcSAG1_NcGRA7_NcROP2 (*** p* < 0.0078, ** p* < 0.0275; Mann–Whitney U test).

**Figure 4 vaccines-11-00156-f004:**
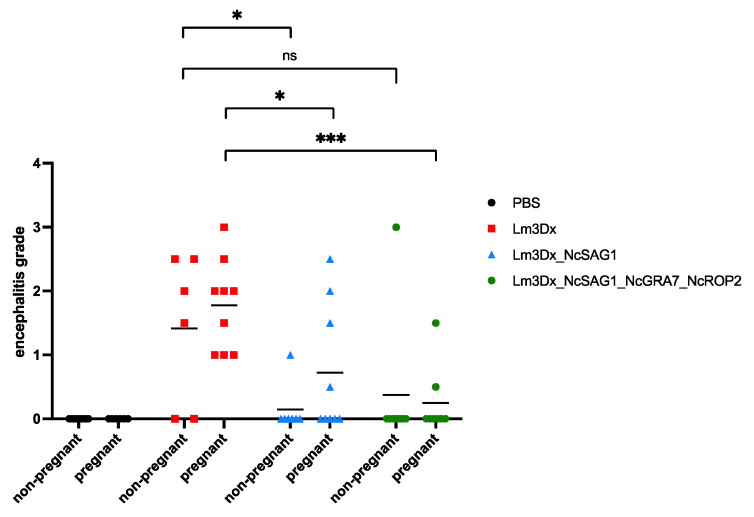
Brain samples were formalin-fixed, embedded in paraffin, cut in slices of 4 µm and stained with HE. *N. caninum*-associated lesions were assessed under the light microscope, and encephalitic grades were scored from 0–3. Grades were analyzed between groups by the non-parametric Kruskal–Wallis test. Afterwards, the encephalitic grades of non-pregnant mice vaccinated with Lm3Dx_NcSAG1 and Lm3Dx_NcSAG1_NcGRA7_NcROP2 were compared to non-pregnant mice that were immunized with the Lm3Dx (vaccination with empty vector and infected with *N. caninum* tachyzoites) using the Mann–Whitney U test (* *p* < 0.0373; ns = not significant). The same statistical analysis was performed comparing dams immunized with Lm3Dx_NcSAG1 and Lm3Dx_NcSAG1_NcGRA7_NcROP2 with dams vaccinated with the empty vector Lm3Dx (* *p* < 0.0313 and *** *p* < 0.0007; Mann–Whitney U test).

**Figure 5 vaccines-11-00156-f005:**
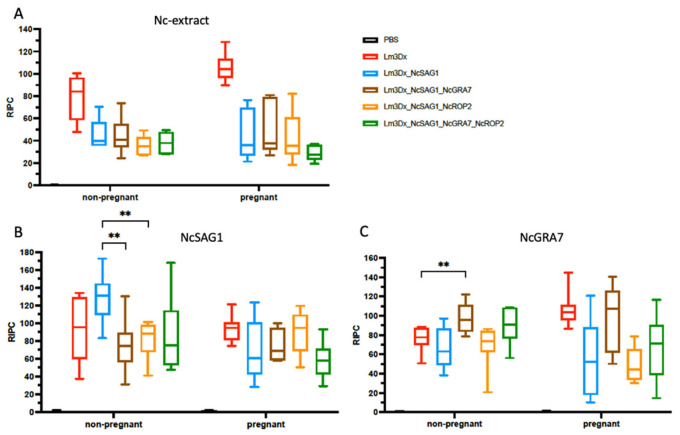
IgG responses of mice measured by ELISA against whole lysates of *N. caninum* tachyzoites (**A**) and against recombinant NcSAG1 (**B**) and NcGRA7 (**C**). Blood was collected at the endpoint (25 days post-partum). ** indicates statistically significant differences with *p* < 0.007.

**Figure 6 vaccines-11-00156-f006:**
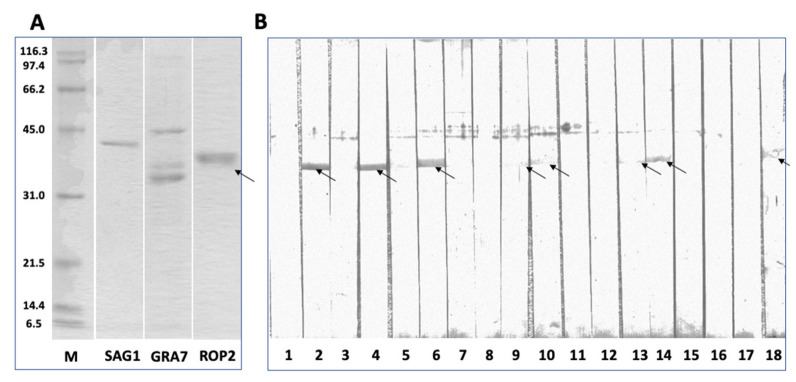
SDS-PAGE of recombinant antigens used for ELISA and Western blot showing the reactivity with recombinant NcROP2 of individual mouse sera from mice vaccinated with the triple mutant Lm3Dx_NcSAG1_NcGRA7_NcROP2. (**A**): the individual recombinant antigens NcSAG1, NcGRA7 and NcROP2 obtained after Ni^2+^ affinity chromatography are shown; M indicates the molecular weights of separated proteins. (**B**): Western blot of recombinant NcROP2. Lane 1 shows a negative control serum; lane 2 a positive control serum; sera in lanes 3–18 are from animals that were vaccinated with the triple mutant and challenged with *N. caninum* tachyzoites; lanes 3–8 are sera from non-pregnant mice; and 9–18 are sera from dams.

**Table 1 vaccines-11-00156-t001:** Experimental groups (*n* = 16) for the evaluation of the mutant *Listeria* vaccine strains.

Group	Vaccine Strain	Vaccine Dose	Infection Strain	Infection Dose
1	Lm3Dx_NcSAG1	1 × 10^7^ CFU *	*N. caninum* Spain-7	1 × 10^5^ tachyzoites
2	Lm3Dx_NcSAG1_NcGRA7	1 × 10^7^ CFU *	*N. caninum* Spain-7	1 × 10^5^ tachyzoites
3	Lm3Dx_NcSAG1_NcROP2	1 × 10^7^ CFU *	*N. caninum* Spain-7	1 × 10^5^ tachyzoites
4	Lm3Dx_NcSAG1_NcGRA7_NcROP2	1 × 10^7^ CFU *	*N. caninum* Spain-7	1 × 10^5^ tachyzoites
5	Lm3Dx (empty vector); C+	1 × 10^7^ CFU *	*N. caninum* Spain-7	1 × 10^5^ tachyzoites
6	Phosphate-buffered saline (PBS); C−	-	BALB/c dermal fibroblasts	1 × 10^5^ cells

* CFU: colony-forming units.

**Table 2 vaccines-11-00156-t002:** Compilation of the results of the vaccine study, including serological analysis, parasite load, litter size and mortality rates of non-pregnant and pregnant mice. The infection control group C+ was immunized with the empty vector Lm3Dx and was challenged as well with *N. caninum* tachyzoites. The animals in the negative control group C− received PBS and were inoculated with 1 × 10^5^ BALB/c dermal fibroblasts. Mice were euthanized 25 days post-partum, and the cerebral parasite load of non-pregnant and pregnant animals as well as of surviving pups was quantified by RT-qPCR. Statistical analysis between animals from the different vaccine groups and the positive control group was performed by the Chi-squared (and Fisher’s exact) test with the GraphPad Prism software. *p*-values lower than 0.05 were considered statistically significant.

Vaccination	Challenge	*N. caninum* Seropositive	NP PCR Brain pos.	Dams PCR Brain pos.	Number of Pups/Dam	Neonatal Mortality ^a^	Postnatal Mortality ^b^	Pups PCR-pos.
Lm3Dx_NcSAG1 1 × 10^7^ CFU	1 × 10^5^ NcSpain-7	16/16	2/7	5/9	61/9 (∅ 6.8)	5/61 (8.2%)	14/56 (25%) ^6^	27/56 (48%) ^6^
Lm3Dx_NcSAG1_NcGRA7 1 × 10^7^ CFU	1 × 10^5^ NcSpain-7	16/16	4/10	3/6 ^3^	40/6 (∅ 6.7)	4/40 (10%)	16/36 (44%) ^6^	19/36 (53%) ^6^
Lm3Dx_NcSAG1_NcROP2 1 × 10^7^ CFU	1 × 10^5^ NcSpain-7	16/16	3/7	2/9 ^4^	57/9 (∅ 6.3)	8/57 (14%)	16/49 (33%) ^6^	22/49 (45%) ^6^
Lm3Dx_NcSAG1_NcGRA7_NcROP2 1 × 10^7^ CFU	1 × 10^5^ NcSpain-7	16/16	1/8 ^1^	2/8 ^4^	47/8 (∅ 5.9)	4/47 (8.5%)	11/43 (25%) ^6^	15/43 (35%) ^6^
C+: Lm3Dx (empty vector) 1 × 10^7^ CFU	1 × 10^5^ NcSpain-7	16/16	6/7	9/9	60/9 (∅ 6.7)	4/60 (6.7%)	52/56 (93%)	53/56 (95%)
C−: phosphate-buffered saline (PBS)	1 × 10^5^ BALB/c DF	0/16	0/10 ^2^	0/6 ^5^	34/6 (∅ 5.7)	2/34 (5.9%)	0/32 (0%) ^6^	0/32 (0%) ^6^

^a^ = numbers of pups born dead or died within the first two days post-partum, ^b^ = number of pups that died between day three post-partum until the end of the study (25 days post-partum), ^1^
*** p* < 0.0087; ^2^ *** *p* < 0.0006; ^3^ * *p* < 0.044; ^4^ ** *p* < 0.0023; ^5^ *** *p* < 0.0002; ^6^ **** *p* < 0.0001.

## Data Availability

No publicly archived datasets for this study are available. All data are presented in this article. For further information, contact the corresponding authors.

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
