# Peer review of "Immunization with a Multivalent Listeria monocytogenes Vaccine Leads to a Strong Reduction in Vertical Transmission and Cerebral Parasite Burden in Pregnant and Non-Pregnant Mice Infected with Neospora caninum"

_vaccines, 2023, doi:10.3390/vaccines11010156_

Round 1

Reviewer 1 Report

Overall, the manuscript complements previous author’s publications regarding the usefulness of Listeria monocitogenes-based vaccines against vertical transmission of Neospora caninum. Their results confirm a reduction in foetal loss, transmission and parasite burden in brain, especially after vaccination with the Lm3Dx_NcSAG1_FRA7_RON2 strain. While vaccine strains seem safe and efficient on reducing foetal transmission, the manuscript does not go at bottom of the immunological mechanisms behind this
efficacy. End-point analysis of cellular immune response (e.g. regulation of Th1/Th2) and/or humoral immune response analysis after immunisation and after challenge would enlighten their results and reinforce the conclusion proposed by the authors regarding the modulation of the innate immune response in the context of the vaccination against neosporosis. Otherwise, a deeper discussion regarding the differential IgG production among groups and their significance during protection against neosporosis would improve the manuscript.

Other minor suggestions include:

- Page 4 line 148; even if generation of Lm3Dx_NcSAG1 strain has already been described, some brief information regarding Listeria strain culture, electroporation and selection features should be added briefly.

- Page 6 line 196; how were bacteria cultured/recovered/treated before inoculation ?

- Page 12 line 388; since parasite load in brains of dams and of pregnant mice is different, figure regarding histological analysis should be also separated

- Page 13 line 411, if all groups were blood sampled before and after challenge, IgG levels should be compared and discussed

- Page 15, line 447, correct to "... sera in lanes 3-18 are from animals that were vaccinated..."

Author Response

We thank reviewer 1 for constructive comments.

Overall, the manuscript complements previous author’s publications regarding the usefulness of Listeria monocitogenes-based vaccines against vertical transmission of Neospora caninum. Their results confirm a reduction in foetal loss, transmission and parasite burden in brain, especially after vaccination with the Lm3Dx_NcSAG1_GRA7_RON2 strain. While vaccine strains seem safe and efficient on reducing foetal transmission, the manuscript does not go at bottom of the immunological mechanisms behind this efficacy. End-point analysis of cellular immune response (e.g. regulation of Th1/Th2) and/or humoral immune response analysis after immunisation and after challenge would enlighten their results and reinforce the conclusion proposed by the authors regarding the modulation of the innate immune response in the context of the vaccination against neosporosis. Otherwise, a deeper discussion regarding the differential IgG production among groups and their significance during protection against neosporosis would improve the manuscript.

Response: We know that this manuscript does not go into the details of cellular and humoral immune responses, as indicated by reviewer 1. However, the primary aim of this work was to establish that this polyvalent vaccine does actually confer increased protection against vertical transmission and increased pup survival, and a more detailed analysis of the cellular and humoral immune responses would require a much larger number of animals. As we have now established increased efficacy, a more detailed study focusing on the potential nature of protective immunity against congenital neosporosis in this model will follow at a later timepoint.

Other minor suggestions include:

- Page 4 line 148; even if generation of Lm3Dx_NcSAG1 strain has already been described, some brief information regarding Listeria strain culture, electroporation and selection features should be added briefly.

Response: We added the following informations (now lane 150-155). Briefly, the JF5203∆actA mutant was used as parental strain to create the vaccine vector JF5203∆actA/inlA/inlB/fosX (Lm3Dx) by in-frame deletion of the three genes by homologous recombination. Lm3Dx_SAG1 was created by electroporating the plasmid pMAD_NactA100AA_SAG1 comprehending the NcSAG1 gene in Lm3Dx. A sequence of selection on plate and temperature changes as described by Arnaud et al. was performed to insert this gene at the locus of the previously removed actA Lm virulence gene.

- Page 6 line 196; how were bacteria cultured/recovered/treated before inoculation?

Response: we have added the following information (now lane 210-215): For each experiment inoculation doses of each vector were prepared as described (19). Briefly, after culturing a colony overnight in 10 ml BHI broth at 37°C, bacteria were washed 3x with sterile PBS and were diluted to achieve a concentration of 1x107 CFU of the respective mutant L. monocytogenes vaccine strain in 50 µl. The concentration was confirmed by plating of the inoculum on BHI-Agar plates (Sigma-Aldrich) and determination of the CFU 24 h after incubation at 37°C.

- Page 12 line 388; since parasite load in brains of dams and of pregnant mice is different, figure regarding histological analysis should be also separated

Response: we have added a modified figure and figure legend (see lanes 418-429)

- Page 13 line 411, if all groups were blood sampled before and after challenge, IgG levels should be compared and discussed

Response: we have only taken blood at the endpoint, not prior to challenge, but introduced a group that was only treated with PBS and was not infected as a control. No IgG was detected in that group (see Figure 5). We did this because our primary aim was to determine efficacy, and blood samples from pregnant animals might have a detrimental influence on pregnancy outcome due to the stress induced by the procedure.

- Page 16, line 473, correct to "... sera in lanes 3-18 are from animals that were vaccinated..."

Response: correction done

Reviewer 2 Report

I find this article very interesting and valuable. Since, there is no effective vaccine against Neospora caninum, the results obtained in this study seems to be very promising.
In their earlier study, based on attenuated Listeria monocytogenes, authors created the mutant Lm3Dx with insert coding the major immunodominant surface antigen NcSAG1.
In this study, authors engineered the Lm3Dx_NcSAG1 vaccine vector to express NcSAG1 simultaneously with NcGRA7 and NcROP2, and triple mutant, NcSAG1, NcGRA7 and NcROP2.
Surface protein NcSAG1, plays an important role in N. caninum host cell invasion. Protein of dense granule, GRA7, plays a role in formation of the parasitophorous vacuole in infected cell. Proteins located in rhoptries are important for the initiation of the host cell invasion processes and the formation of the parasitophorous vacuole membrane.
All selected proteins are critical for the parasite invasion and survival in the host’s cells.  
Using double and triple multivalent Lm3Dx vaccine authors challenged vaccinated and pregnant mice with N. caninum tachyzoites to assess vaccine-mediated protection.

The results obtained here are very promising. Immunization with Lm3Dx_NcSAG1_NcGRA7_NcROP2 resulted in a strong reduction of vertical transmission and in an increased postnatal pup survival, up to 70%.

I have some questions to authors and comments.
In the chapter Introduction, in lines 124-135 the text needs some editorial corrections, (font).
In chapter Results, in description of Figure 6, in line 447, there is “sera in lanes 2-18 are from animals that were vaccinated with the triple 447 mutant and challenged with N. caninum tachyzoites” but should be “sera in lines 3-18”, I believe.

I recommend this article to be published in Vaccines.

Author Response

Many thanks to reviewer 2 for the positive evaluation of our manuscript.

I find this article very interesting and valuable. Since, there is no effective vaccine against Neospora caninum, the results obtained in this study seems to be very promising.
In their earlier study, based on attenuated Listeria monocytogenes, authors created the mutant Lm3Dx with insert coding the major immunodominant surface antigen NcSAG1. 
In this study, authors engineered the Lm3Dx_NcSAG1 vaccine vector to express NcSAG1 simultaneously with NcGRA7 and NcROP2, and triple mutant, NcSAG1, NcGRA7 and NcROP2.
Surface protein NcSAG1, plays an important role in N. caninum host cell invasion. Protein of dense granule, GRA7, plays a role in formation of the parasitophorous vacuole in infected cell. Proteins located in rhoptries are important for the initiation of the host cell invasion processes and the formation of the parasitophorous vacuole membrane.
All selected proteins are critical for the parasite invasion and survival in the host’s cells.  
Using double and triple multivalent Lm3Dx vaccine authors challenged vaccinated and pregnant mice with N. caninum tachyzoites to assess vaccine-mediated protection.

The results obtained here are very promising. Immunization with Lm3Dx_NcSAG1_NcGRA7_NcROP2 resulted in a strong reduction of vertical transmission and in an increased postnatal pup survival, up to 70%.

I have some questions to authors and comments.

In the chapter Introduction, in lines 124-135 the text needs some editorial corrections, (font).

Response: corrections were made

In chapter Results, in description of Figure 6, in line 473, there is “sera in lanes 2-18 are from animals that were vaccinated with the triple mutant and challenged with N. caninum tachyzoites” but should be “sera in lines 3-18”, I believe.

Response: corrections were made

I recommend this article to be published in Vaccines.